# Gut Microbiota Changes by an SGLT2 Inhibitor, Luseogliflozin, Alters Metabolites Compared with Those in a Low Carbohydrate Diet in db/db Mice

**DOI:** 10.3390/nu14173531

**Published:** 2022-08-27

**Authors:** Shinnosuke Hata, Takuro Okamura, Ayaka Kobayashi, Ryo Bamba, Tomoki Miyoshi, Hanako Nakajima, Nobuko Kitagawa, Yoshitaka Hashimoto, Saori Majima, Takafumi Senmaru, Hiroshi Okada, Emi Ushigome, Naoko Nakanishi, Hiroshi Takakuwa, Ryoichi Sasano, Masahide Hamaguchi, Michiaki Fukui

**Affiliations:** 1Department of Endocrinology and Metabolism, Graduate School of Medical Science, Kyoto Prefectural University of Medicine, Kyoto 602-8566, Japan; 2Department of Endocrinology, Diabetes and Metabolism, Kyoto Chubu Medical Center, Nantan 629-0197, Japan; 3Department of Diabetology, Matsushita Memorial Hospital, Moriguchi 570-8540, Japan; 4Agilent Technologies, Chromatography Mass Spectrometry Sales Department, Life Science and Applied Markets Group, Hachioji 192-0033, Japan; 5AiSTI Science Co., Ltd., Wakayama 640-8390, Japan

**Keywords:** sodium-glucose co-transporter-2 inhibitor, luseogliflozin, sarcopenia, sarcopenic obesity, metabolites, gut microbiota

## Abstract

In recent years, sarcopenic obesity has been considered central pathological factors in diabetes. This study aimed to compare the effect of luseogliflozin, a sodium-glucose co-transporter-2 inhibitor (SGLT2i), on sarcopenic obesity in comparison to that of a low-carbohydrate diet (LCD). Twenty-week-old male db/db mice were fed a normal diet (Ctrl), LCD, and normal diet with 0.01% *w*/*w* luseogliflozin (SGLT2i) for eight weeks. Skeletal muscle mass and grip strength decreased in the LCD group mice compared to those in the control group, while they increased in the SGLT2i group mice. The amino acid content in the liver, skeletal muscle, and serum were lower in the LCD group than those in the Ctrl group but increased in the SGLT2i group mice. Short-chain fatty acids in rectal feces were lower in the LCD group mice than those in the Ctrl group, whereas they were higher in the SGLT2i group mice. The abundance of *Gammaproteobacteria*, *Enterobacteriaceae*, *Escherichia*, *Enterobacterales*, and *Bacteroides caccae* species increased in the LCD group compared to the other two groups, whereas the abundance of *Syntrophothermus lipocalidus*, *Syntrophomonadaceae* family, *Parabacteroidesdistasonis distasonis*, and the genus *Anaerotignum* increased in the SGLT2i group. Luseogliflozin could prevent sarcopenic obesity by improving amino acid metabolism.

## 1. Introduction

The prevalence of diabetes mellitus has increased rapidly, and preventing the progression of sarcopenia, a diabetic complication, has become imperative to decrease the number of bedridden patients. In recent years, sarcopenic obesity, a decrease in skeletal muscle mass along with an increase in body fat, has been assumed to be the main pathogenesis of diabetes-associated obesity, and changes in skeletal muscle mass are associated with fatty liver and glycemic control [1]. Therefore, innovative animal models are expected to elucidate the pathomechanisms of sarcopenic obesity and lead to novel therapeutic approaches.

Our previous study demonstrated sarcopenic obesity in db/db mice and found elevated levels of Foxo1 expression and saturated fatty acids in the skeletal muscle [2,3]. In these studies, while the concentrations of saturated fatty acids including palmitic, myristic, stearic, and lauric acid in the skeletal muscles of sarcopenic obese mice are elevated, the concentrations of these saturated fatty acids are significantly reduced in the skeletal muscles of mice administered with luseogliflozin, a sodium-glucose co-transporter-2 inhibitor (SGLT2i), and increased muscle strength and muscle mass are observed. In addition, the expression of fatty acid transporters, including CD36, in the skeletal muscle is reduced in association with a decrease in blood glucose. Thus, we revealed that luseogliflozin improves the serum lipidome in addition to blood glucose. Although the direct mechanism of regulating gene expression by improving intracellular metabolic conditions has not yet been elucidated, our previous study suggested that luseogliflozin-mediated reduction in saturated fatty acid concentrations in the skeletal muscle involves suppressing the expression of Foxo1 and other skeletal muscle atrophy-related genes.

Dietary treatment for sarcopenic obesity is limited; however, protein [4], calcium, and vitamin D supplements [5] are potentially effective against sarcopenic obesity. Many clinical trials have evaluated the diet for obesity and type 2 diabetes, with low-carbohydrate diets (LCDs) demonstrating effectiveness, although they are still controversial. LCDs are classified into the following four major categories based on their carbohydrate/energy ratio: ketogenic diets, energy from carbohydrates (%): <10%, low carbohydrates: <26%, moderate carbohydrates: 26–45%; and high carbohydrates: 45–55% [6]. A systematic review and meta-analysis of randomized controlled studies examining the effects of LCD suggested that reducing dietary carbohydrates would clinically improve the management of type 2 diabetes. It was also suggested that a considerable LCD (<50 g/day) may hinder dietary adherence [7]. Additionally, in another meta-analysis of over 400,000 individuals, both high (>70%) and low (<40%) carbohydrate intake was associated with an increased risk of mortality than moderate carbohydrate (50–55%) intake [8]. In summary, LCDs are beneficial for the short-term reduction of body weight; however, the longer-term effectiveness of LCDs remains controversial [9]. Furthermore, the effects of LCD on sarcopenia remain controversial. In humans, LCD was reported to prevent muscle protein catabolism under energy restriction, when it includes sufficient amounts of protein [10], whereas Nakao et al. [11]. described that LCDs significantly increased the mRNA levels of oxidative stress-responsive genes, including Sod1, in the skeletal muscle and suppress muscle protein synthesis in a mouse study. Therefore, the efficacy of LCDs for sarcopenic obesity, a combination of obesity and sarcopenia, remains unclear.

Furthermore, carbohydrate restriction due to LCD may consequently restrict fiber intake. According to the findings of several human studies, LCD decreases short-chain fatty acid (SCFA) levels in feces and causes dysbiosis [12,13,14]. The equilibrium between obligate and facultative anaerobes is considered important, and a disruption in this equilibrium is called dysbiosis [15]. Biophilic anaerobes are fermentative bacteria, which mainly use dietary fiber as a nutrient source to produce SCFAs. The SCFAs further stimulate intestinal resident type 3 innate lymphocytes via FFAR3 to release interleukin (IL)-22 [16]. IL-22 thickens the mucin layer, strengthens the intestinal barrier, and prevents endotoxins and lipopolysaccharides from entering the body [17]. Through these mechanisms, IL-22 has been shown to protect against the onset of metabolic syndrome; thus, LCD may induce intestinal inflammation by reducing the promotion of SCFAs in the intestinal tract.

In a single dose luseogliflozin (3 mg/kg) study conducted in db/db mice, urinary glucose excretion levels at 8 h after administration was 408 mg, compared with 228 mg in the pathological control group (Taisho Pharmaceutical in-house data, common technical document 2.6.2.2.2.2). The differences in urinary glucose excretion levels between the control group and luseogliflozin in db/db mice were 100 mg at 8h [18], and 200 mg at 6h [19]. Therefore, we estimated the maximum urinary glucose excretion effect of luseogliflozin (0.01%) on 8-week-old db/db mice in a previous study to be 800 mg/day (300–800 mg/day). If the anti-sarcopenic effect of luseogliflozin is solely due to urinary glucose excretion, it is assumed that this effect would be exceeded by the anti-sarcopenic effect of a carbohydrate-restricted diet. Since the daily carbohydrate intake was 6.0 g in the study, we prepared a 51% carbohydrate (LCD 51%, D20031902) diet, with a daily intake of 4.4 g, a reduction of 1600 mg carbohydrate, which is twice the urinary glucose excretion effect of 800 mg/day.

The purpose of this study was to confirm the anti-sarcopenic effect of luseogliflozin on LCD, using db/db mice fed an equal weight dose diet of AIN93G with either 51% LCD (64% carbohydrate ratio) or 0.01% luseogliflozin.

## 2. Materials and Methods

### 2.1. Mice and Experimental Design

Approval by the Committee for Animal Research of the Kyoto Prefectural University of Medicine was obtained for all experimental protocols. Iar-Leprdb/Leprdb mice (db/db mice) were housed in the Kyoto Prefectural University of Medicine Animal Facility (M2020-38). We obtained male homozygous db/db mice with diabetes at 20 weeks of age from Shimizu Laboratory Supplies (Kyoto, Japan). The mice were given a standard diet (4 kcal/g, carbohydrate kcal 64%, fat kcal 16%; New Brunswick, NJ, USA) or a low-carbohydrate diet (4 kcal/g, carbohydrate kcal 51%, fat kcal 21%; Research Diets Inc., New Brunswick, NJ, USA) (Table 1) for 8 weeks from 20 weeks of age. They were separated into the three groups described below. We performed the following experiments with 6 mice per group: (i) db/db mice fed a normal diet (6.0 g/day) without SGLT2i (Ctrl), (ii) db/db mice fed an LCD (4.4 g/day), and (iii) db/db mice fed a normal diet (6.0 g/day) with luseogliflozin (SGLT2i). We used luseogliflozin [Lusefi^®^, Tokyo, Japan)], an oral hypoglycemic agent and second-generation SGLT2i originally developed by Taisho Pharmaceutical Co., Ltd. in this study. Luseogliflozin was added to the diet at a ratio of 0.01% per weight, which was a similar dosage to our previous study [2,3]. The effect of the 0.01% luseogliflozin mixed diet was set at a urinary glucose excretion of 300–800 mg/day. At the age of 28 weeks, we euthanized all mice by administration of the following combination of anesthetics: butorphanol (5.0 mg/kg), midazolam (4.0 mg/kg), and medetomidine (0.3 mg/kg) (Figure 1A). The weight of soleus muscle, plantaris muscle, liver, and epididymal fat were measured.

### 2.2. Fasting Blood Glucose Measurement and Glucose Tolerance Tests

Blood glucose levels after 14 h fasting were checked at 20, 22, 24, 26, and 28 weeks. The intraperitoneal glucose tolerance test (iPGTT) (1 mg/g) (6 mice in each group, two days before euthanasia) and insulin tolerance test (ITT) (0.5 U/kg) (three days before euthanasia) were evaluated at 28 weeks of age, after 5 and 14 h of fasting, respectively. Then, the area under the curve (AUC) was calculated. We measured blood glucose levels with a glucometer (Glutestmint II; Sanwa Kagaku Kenkyusho, Nagoya, Japan) in the tail vein. The data of ITT and iPGTT were calculated and analyzed by obtaining the AUC.

### 2.3. Grip Strength Assay

Grip strength was assessed with a mouse grip strength meter (model DS2-50N, IMADA Co., Ltd., Toyohashi, Japan) in another batch of 28-week-old mice. Six consecutive measurements were taken at 1-min intervals. The researchers were blinded to the group allocation of the mice.

### 2.4. Amino Acid Quantification in Sera, Liver and Plantaris Muscle, and SCFA Determination in Feces

The amino acid composition of sera, liver, and the plantaris muscle, as well as the SCFA composition of rectal feces, were determined with gas chromatography (GC)–mass spectrometry (MS) performed using an Agilent 7890B/7000D System (Agilent Technologies, Santa Clara, CA, USA). Samples of serum (50 μL), liver (20 mg) rectal feces (20 mg), and plantaris muscle (20 mg) were mixed with 500 μL diluted water and 500 μL acetonitrile and, ground in a ball mill at 4000 rpm for two minutes. Further, the specimens were shaken at 1000 rpm and 37 °C for thirty minutes and centrifuged at 14,000 rpm and room temperature for three minutes. The supernatant (500 μL) was separated, then added to 500 μL acetonitrile, and shaken at 1000 rpm and 37 °C for three minutes. The specimens were centrifuged at 14,000 rpm and room temperature for three minutes, and the pH of the supernatant was controlled to eight with 0.1 mol/L NaOH, to extract the SCFAs.

SCFA and amino acid concentrations were measured using GC/MS with online solid-phase extraction (SPE). In the SPE-GC system SGI-M100 (AiSTI Science, Wakayama, Japan), SPE and injection into the GC/MS system were automated after placing vials of the sample on an autosampler tray. We used flash-SPE ACXs (AiSTI Science) for solid-phase stratification. Aliquots (50 µL) of each sample extract were obtained, loaded onto the solid phase, and washed with water and acetonitrile (1:1). Subsequently, the samples were dehydrated with acetone, impregnated with 4 μL N-tert-butyldimethylsilyl-N-methyltrifluoroacetamide-toluene solution (1:3), and eluted with hexane after derivatization in the solid phase. The final product was injected using a programmed temperature vaporizer injector, LVI-S250 (AiSTI SCIENCE), with temperature maintained at 150 °C for 0.5 min, increased gradually at a rate of 25 °C/min up to 290 °C, and then maintained for 16 min. The samples were loaded onto a capillary column, Vf-5ms [30 m (length) × 0.25 μm (membrane thickness) × 0.25 mm (inner diameter); Agilent Technologies]. The column temperature was maintained at 60 °C for three minutes, increased at a rate of 10 °C/min to 100 °C, increased subsequently at a rate of 20 °C/min to 310 °C, and maintained at 310 °C for seven minutes. The sample was injected in the split mode at a ratio of 20:1. The SCFAs were detected in the scan mode (m/z:70–470). Each result was normalized to the peak height of norleucine (0.02 nmol/μL) for amino acids and tetradeuteroacetic acid (0.02 nmol/μL) for SCFAs [20].

### 2.5. Quantification of Free Fatty Acids in the Soleus Muscle

The soleus muscle composition of free fatty acids was determined by using the same gas chromatography-mass spectrometry (GC/MS) system as that used for SCFA and amino acid quantification. and an Agilent 7890B/7000D (Agilent Technologies, Santa Clara, CA, USA). Soleus muscle (15 µg) was methylated with a fatty acid methylation kit (Nacalai Tesque, Kyoto, Japan). The final product was loaded onto a Varian capillary column (DB-FATWAX UI; Agilent Technologies). The capillary column used was CP-Sil 88 for FAME (length, 100 mm; membrane thickness, 0.20 μm; inner diameter, 0.25 mm; Agilent Technologies) for fatty acid separation. The column temperature was maintained at 100 °C for four minutes, increased gradually at a rate of 3 °C/min to 240 °C, and maintained for seven minutes. The samples were injected in the split mode at a ratio of 5:1. The fatty acid methyl esters were detected in the selected ion monitoring mode. Each result was normalized to the peak height of the C17:0 internal standard.

### 2.6. Gene Expression in the Soleus Muscle

The soleus muscle of fasting mice was resected, immediately frozen in liquid nitrogen, homogenized in ice-cold QIAzol Lysis reagent, and total RNA was isolated as described in the manufacturer’s recommendations. Total RNA (0.5 µg) was reverse-transcribed using a High-Capacity cDNA Reverse Transcription Kit (Applied Biosystems, Foster City, CA, USA) for first-strand cDNA synthesis with an oligonucleotide dT primer and random hexamer priming, as instructed by the manufacturer. We performed the reverse transcription (RT) reaction at 37 °C for 120 min, and the inactivation of RT at 85 °C for five minutes. The mRNA expression levels of Foxo1, Trim63, Fbxo32, Hdac4, Il-6, Il-1β, and Scd1 were quantified with real-time reverse transcription polymerase chain reaction (RT-PCR). The relative expression levels of genes were normalized to the gapdh cycle threshold (CT) values and quantified using the comparative threshold cycle 2^−ΔΔCT^ method, as previously described. Signals from db/db mice without SGLT2i were assigned a relative value of 1.0. We performed RT-PCR with TaqMan Fast Advanced Master Mix (Applied Biosystems), by following the manufacturer’s instructions. The following PCR conditions were applied: 1 cycle at 50 °C for two minutes and at 95 °C for twenty seconds, followed by 40 cycles of one second at 95 °C, and s0 s at 60 °C.

### 2.7. Metagenomic Analysis

Fecal samples were obtained from the appendix and collected in a cryotube. Immediately following this, the samples were attached to liquid nitrogen for cryopreservation and stored in liquid nitrogen until DNA extraction. We collected three fecal samples one at a time from the appendix of three mice, excluding one large and one small mouse in a cage of each group. Microbial DNA was obtained from frozen fecal samples with the QIAamp^®^ DNA Stool Mini Kit (Qiagen, Venlo, The Netherlands) according to the manufacturer’s instructions [21].

We performed whole-genome shotgun sequencing by using a HiSeq 2000/2500/4000 system (Illumina) at the Bioengineering Lab. Co., Ltd., Sagamihara, Japan).

### 2.8. Statistical Analyses

We analyzed data with the JMP software (version 14.0; SAS, Cary, NC, USA). Differences between two groups were evaluated using the unpaired t-test, and differences among more than three groups were evaluated with Tukey’s multiple comparison test. Statistical significance was defined at *p* < 0.05. Each figure was created using the GraphPad Prism software (version 9.0; San Diego, CA, USA).

## 3. Results

### 3.1. Effect of SGLT2i on Body Weight and Glucose Homeostasis

The body weight of the LCD group mice was lower than that of the Ctrl and SGLT2i groups after the 8-week treatment (28 weeks: Ctrl, *p* = 0.002; SGLT2i, *p* < 0.0001) (Figure 1B). The fasting blood glucose levels in the SGLT2i group tended to be lower than those in the other two groups after the age of 22 weeks (Figure 1C). Compared to the Ctrl group and LCD group, the AUCs of ITT and iPGTT in the SGLT2i group were lower (Figure 1D–G).

### 3.2. Effect of SGLT2i on Muscle Strength

Grip strength and the grip strength/body weight ratio measured at 28 weeks of age were both significantly higher in the SGLT2i mice compared to those in the other two groups (Figure 1H,I).

### 3.3. Effect of SGLT2i on Visceral Fat and Skeletal Muscle

The weight and organ weight/body weight ratios were examined. A significant decrease in epididymal fat was observed in the SGLT2i group compared with that in the other two groups, whereas the LCD group showed a significant increase in epididymal fat compared with that in the other two groups (Figure 2A,B). Soleus muscle weight and soleus muscle weight/body weight ratio (×1000) and plantaris muscle weight and plantaris muscle weight/body weight ratio (×1000) were not significantly different between the Ctrl and LCD groups but were significantly higher in the SGLT2i group than those in the Ctrl and LCD groups (Figure 2C–F). The cross-sectional areas of the soleus and plantaris muscles was decreased in the LCD group and increased in the SGLT2i group, compared to the Ctrl group (Figure 2G,H).

### 3.4. Effect of SGLT2i on Amino Acid Concentrations in the Sera, Liver, and Skeletal Muscle

Further, branched-chain amino acid concentrations in the serum, liver, and skeletal muscle were investigated. The levels of valine, leucine, and isoleucine concentrations in the serum, liver, and skeletal muscle of the SGLT2i mice were significantly higher than those in the Ctrl and LCD mice (Figure 3A–I). Leucine concentrations tended to be lower in the LCD mice than that in the Ctrl mice, and the concentrations of valine and isoleucine in the skeletal muscle were significantly lower in the LCD mice than those in the Ctrl mice.

### 3.5. Effect of SGLT2i on SCFAs in Feces and Saturated Fatty Acids in the Skeletal Muscle

The LCD mice had significantly lower fecal concentrations of SCFAs including acetic acid, propanoic acid, and butanoic acid than the Ctrl mice, while those in the SGLT2i group were higher than those in the Ctrl group and LCD group (Figure 4A–C).

The concentrations of saturated fatty acids, including lauric, myristic, palmitic, and stearic acid, in the skeletal muscle of the LCD group, were higher than those in the skeletal muscle of the Ctrl group, whereas those of the SGLT2i group were lower than those of the Ctrl and LCD groups (Figure 4D–G). In contrast, the concentration of the unsaturated fatty acid, oleic acid, in the skeletal muscle of the SGLT2i group was higher than that of the Ctrl group, while that of the LCD group was lower than that of the Ctrl group (Figure 4H).

### 3.6. SGLT2i-Mediated Changes in the Gene Expression Levels Involved in Muscle Atrophy, Inflammation, and Fatty Acid Synthesis in the Skeletal Muscle

Further, the gene expression levels involved in muscle atrophy, inflammation, and fatty acid synthesis were investigated using RT-PCR. The gene expression levels related to muscle loss, such as *Foxo1* and *Fbxo32*, in the LCD mice, was higher than that in the Ctrl mice. The gene expression levels of *Foxo1*, *Trim63*, *Fbxo32*, and *Hdac4* in the SGLT2i group was lower than that in the LCD mice (Figure 5A–D). In addition, the gene expression involved in inflammation, such as *Il-6* and *Il-1 beta*, was higher in the LCD mice than that in the Ctrl and SGLT2i groups (Figure 5E,F). Similarly, the gene expression levels involved in fatty acid synthesis, such as *Scd1*, in the LCD group was higher than that in the Ctrl and SGLT2i mice (Figure 5G).

### 3.7. Changes in Gut Microbiota by SGLT2i

Shotgun metagenomic analysis of appendicular feces was performed, and LEfSe analysis was performed to compare the gut microbiota among the three groups (Figure 6). In the LCD group, the abundance of bacteria from the class *Gammaproteobacteria*, the family Enterobacteriaceae, the order *Enterobacterales*, and the species *Bacteroides caccae* increased compared to that in the other two groups. In the SGLT2i group, the abundance of species *Syntrophothermus lipocalidus*, the family *Syntrophomonadaceae*, the species *Parabacteroides distasonis*, and the genus *Anaerotignum* increased.

## 4. Discussion

In the present study, compared to the normal diet (Ctrl) group and the luseogliflozin (SGLT2i) administered group, the LCD group showed significantly worse glucose tolerance, decreased skeletal muscle mass, decreased muscle strength, increased saturated fatty acids, decreased amino acids in the blood and skeletal muscle, and altered gut microbiota. In contrast, compared to the animals in the Ctrl and LCD groups, mice in the SGLT2i group, who were expected to excrete glucose in the urine equivalent to the glucose restriction in LCD, showed significantly improved glucose tolerance, increased skeletal muscle mass, increased muscle strength, decreased saturated fatty acids, and increased amino acids in the blood and skeletal muscle.

We performed a shotgun metagenomic analysis of the gut microbiota, and LEfSe analysis demonstrated that bacterial abundance of the class *Gammaproteobacteria*, which increased in the LCD group, is increased in obese and healthy human participants [22]. In addition, an increase in the bacterial abundance of the family *Enterobacteriaceae* has been reported by lipopolysaccharides and exhibits high endotoxin activity [23]. The bacterial abundance of the order *Enterobacterales* is increased in the intestines of adults with severe obesity and diabetes [24]. Increased abundance of the species *Bacteroides caccae* was reported in the intestines of diabetic patients [25]. In contrast, the SGLT2i group showed a significant increase in the abundance of the species *Syntrophothermus lipocalidus*, family *Syntrophomonadaceae*, and genus *Anaerotignum*, which are involved in the biosynthesis of SCFAs such as acetic acid, propionic acid, and butyric acid [26,27,28]. In addition, the species *Parabacteroides distasonis*, which is abundant in the intestines of patients on low-calorie ketogenic diets or undergoing sleeve gastrectomy bariatric surgery, was significantly increased in the SGLT2i group [29]; this species is also negatively correlated with obesity and metabolic syndrome [30]. The correlation between SGLT2i and modification of the gut microbiota is well established [31,32,33] however, most of these studies are focused on marker-based amplicon sequencing, such as 16s rRNA genes. The deep shotgun sequencing performed in this study allowed for a more detailed analysis of the gut microbiota. In this context, to our knowledge, it was observed for the first time that the administration of luseogliflozin results in the proliferation of gut microbiota involved in SCFA biosynthesis. However, the mechanism by which SGLT2i administration modifies gut microbiota is still unknown. We hypothesized that the inhibition of SGLT2 by luseogliflozin was partly responsible for the altered intestinal metabolites by reducing the absorption of simple sugars; further studies are needed to substantiate this hypothesis. The LCD group showed dysbiosis caused by a decrease in carbohydrates. This was thought to be mainly due to a decrease in fiber intake, owing to a decrease in carbohydrates (Corn starch intake of 2385 mg/day:57.24 mg/day of fiber in the Ctrl and SGLT2i groups, and 786 mg/day:18.86 mg/day of fiber in the LCD group). Furthermore, this was considered a mechanism for improving metabolic disorders specific to luseogliflozin, which cannot be obtained with an LCD.

SCFAs (acetic, propionic, and butyric) in rectal feces are representative enterobacteria-fermented metabolites, derived from dietary fiber substrates. At the molecular level, they contribute closely to the homeostasis of living organisms, by being involved in the energy metabolism of the host, influencing immune function, and epigenomic regulation. Intestinal SCFAs act on pancreatic beta cells via the bloodstream to promote insulin secretion and improve insulin resistance in adipocytes, thereby contributing to the prevention and treatment of metabolic diseases [34,35]. Increased SCFAs in the intestine enhance the intestinal barrier function by increasing the mucin layer of the intestinal mucosa via GPR43, which is expressed on type 3 innate lymphocytes in the intrinsic layer of the small intestinal mucosa [36]. One of the mechanisms of action of the anti-sarcopenic obesity effect of luseogliflozin is the reduction of saturated fatty acid concentrations in the skeletal muscle by improving systemic lipid metabolism [3]. In that study, we found that administration of palmitic acid to C2C12 myotube cells promoted the expression of fatty acid synthesis-related genes including *Scd1*, skeletal muscle atrophy-related genes, and inflammation-related genes. The improvement of fatty acid metabolism as described above may contribute to the decreased expression of atrophy- and inflammation-related genes in skeletal muscle by administration of SGLT2 inhibitors in this study. Simultaneously, the improvement of amino acid metabolism throughout the body via changes in the intestinal microflora could be also one of the mechanisms underlying the decreased expression of inflammation-related genes. We demonstrated that luseogliflozin improved amino acid metabolism, resulting in an increase in amino acid concentrations in the skeletal muscle in the present study. Although it is known that SGLT2i improves amino acid metabolism by improving mitochondrial function [37,38], to our knowledge, this is the first study to quantify amino acid concentrations in the serum, liver, and skeletal muscle and to prove that SGLT2i causes an increase in amino acid concentration. Insulin signaling is also essential for metabolism and amino acid transport in skeletal muscles and is known to play an important role in regulating muscle protein synthesis [39,40]. The increased amino acid concentrations in the skeletal muscle of the SGLT2i mice may be due to the fact that SCFAs in the gut of the SGLT2i mice activate ILC3 in the mucosal intrinsic layer of the small intestine, enhancing the protective effect of IL-22 released by ILC3 on mucosa and reducing the influx of inflammatory substances such as lipopolysaccharides and endotoxins in the body, thereby reducing systemic inflammation and inflammation in the skeletal muscle. Inflammation within skeletal muscles increases muscle insulin resistance [41]. It is suggested that reduced muscle insulin resistance could enhance amino acid absorption into the skeletal muscles, as described here. Further studies are needed to test this hypothesis, including the expression level of ILC3 in the small intestine, measurement of short-chain fatty acid levels in the intestine, and measurement of amino acid levels in the skeletal muscle in IL-22 knockout mice. We analyzed the expression of genes involved in muscle atrophy, inflammation, and fatty acid synthesis by RT-PCR, whereas protein expression was not examined in this study.

In summary, to our knowledge, this study revealed LCD-induced dysbiosis for the first time, using db/db mice. Luseogliflozin treatment was found to increase the abundance of intestinal bacteria involved in the synthesis of SCFAs, leading to improved amino acid metabolism. Thus, through these mechanisms, luseogliflozin could successfully prevent sarcopenic obesity.

## Figures and Tables

**Figure 1 nutrients-14-03531-f001:**
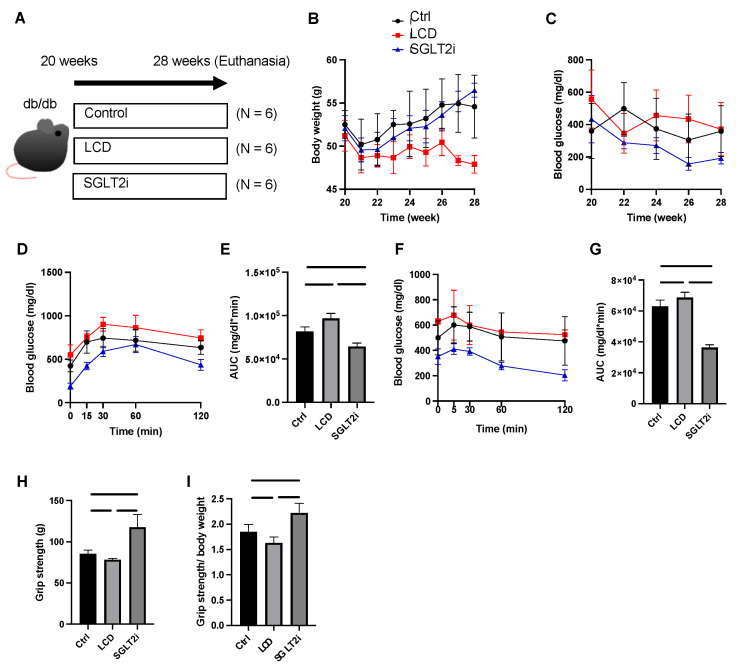
Low-carbohydrate diet (LCD) aggravated glucose impairment and reduced muscle strength, whereas sodium-glucose co-transporter-2 inhibitor (SGLT2i) improved them. (**A**) Overview of the feeding and euthanasia protocol. (**B**) Body weight changes. (**C**) Fasting blood glucose changes. (**D**,**E**) Intraperitoneal glucose tolerance test (iPGTT) results and the area under the curve (AUC) of iPGTT. (**F**,**G**) Insulin tolerance test (ITT) results and the AUC of ITT. Luseogliflozin improved their impaired glucose tolerance. (**H**,**I**) Absolute and relative grip strength. Data are mean ± standard deviation (SD). One-way analysis of variance (ANOVA) was conducted for statistical analysis.

**Figure 2 nutrients-14-03531-f002:**
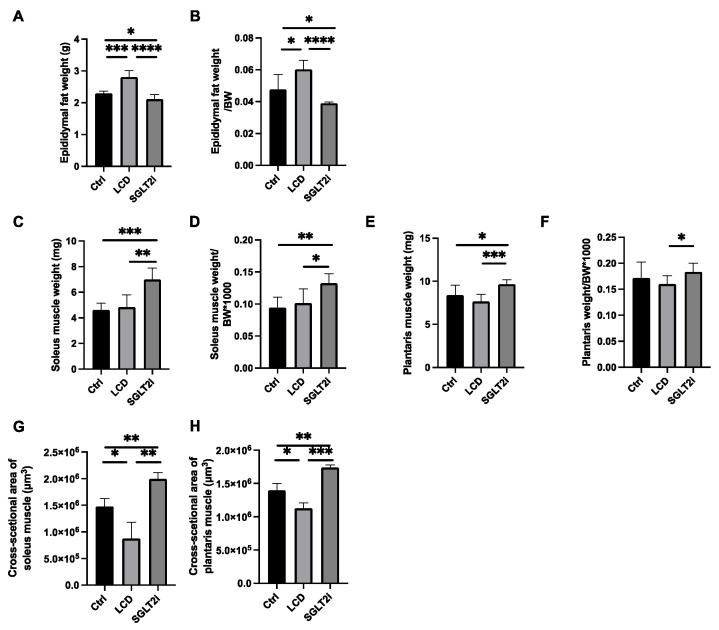
LCD increased visceral fat weight and decreased skeletal muscle, whereas SGLT2i decreased visceral fat weight and increased skeletal muscle. (**A**,**B**) Absolute and relative epididymal fat weights. (**C**,**D**) Absolute and relative soleus muscle weights. (**E**,**F**) Absolute and relative plantaris muscle weights. (**G**,**H**) Cross-sectional areas of the soleus and plantaris muscles. Data are mean ± SD. One-way ANOVA was conducted for statistical analysis. * *p* < 0.05, ** *p* < 0.01, *** *p *< 0.001, and **** *p* < 0.0001.

**Figure 3 nutrients-14-03531-f003:**
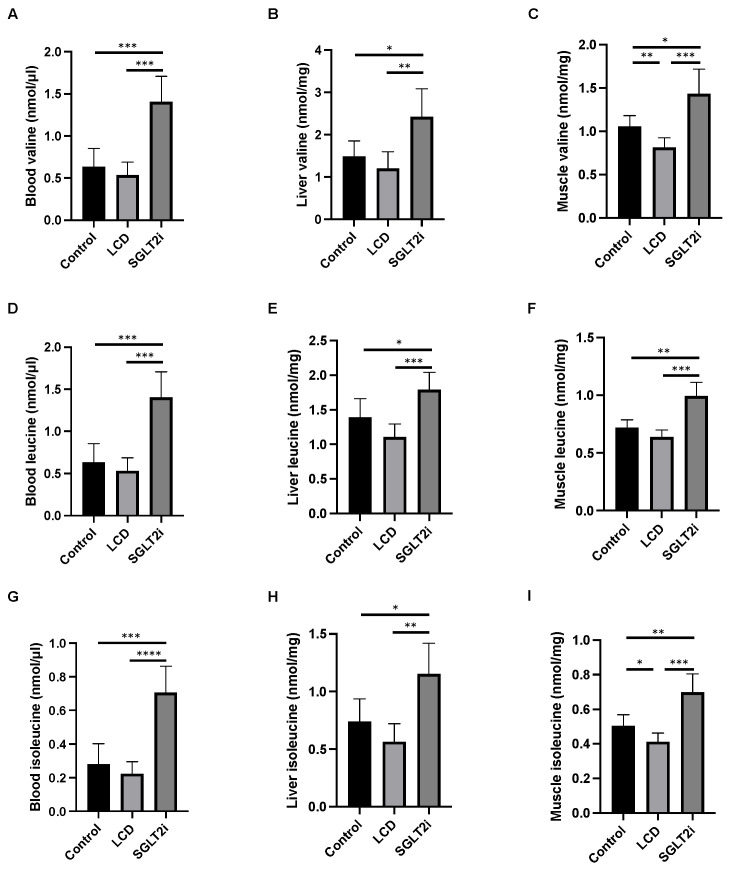
The amino acid concentrations in the sera, liver, and muscle. Concentration of valine in (**A**) serum, (**B**) liver, and (**C**) muscle. Concentration of leucine in (**D**) serum, (**E**) liver, and (**F**) muscle. Concentration of isoleucine in (**G**) serum, (**H**) liver, and (**I**) muscle. Data are mean ± SD. Statistical analyses were conducted with ANOVA. * *p* < 0.05, ** *p* < 0.01, *** *p *< 0.001, and **** *p* < 0.0001.

**Figure 4 nutrients-14-03531-f004:**
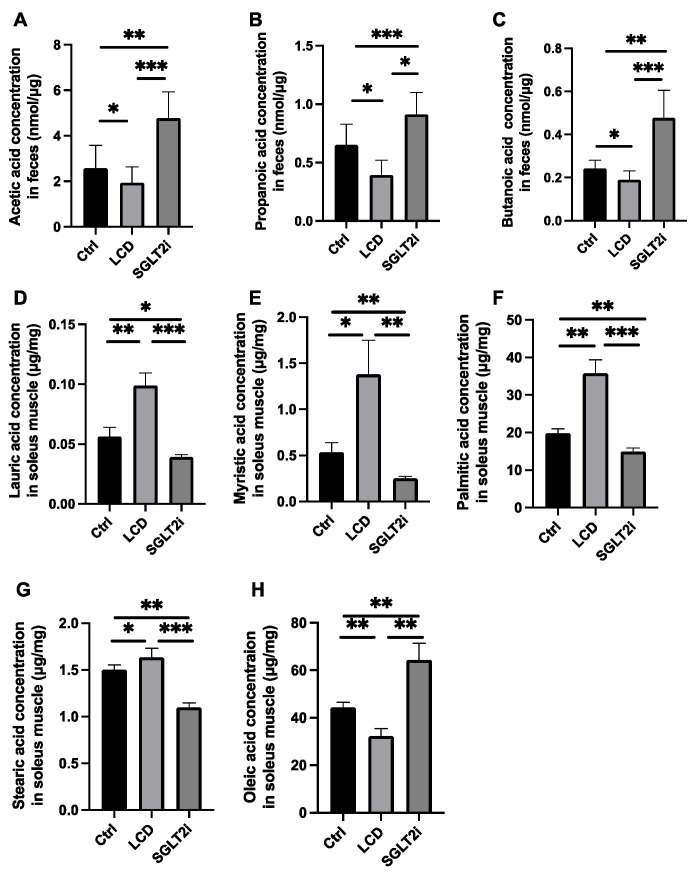
The fecal concentration of short-chain fatty acid and the concentration of saturated fatty acid and unsaturated fatty acids in the soleus muscle. The fecal concentration of (**A**) acetic acid, (**B**) propanoic acid, and (**C**) butanoic acid. The concentrations of (**D**) lauric acid, (**E**) myristic acid, (**F**) palmitic acid, (**G**) stearic acid, and (**H**) oleic acid in the soleus muscle. Data are mean ± SD. Statistical analyses were conducted using ANOVA. * *p* < 0.05, ** *p* < 0.01, and *** *p *< 0.001.

**Figure 5 nutrients-14-03531-f005:**
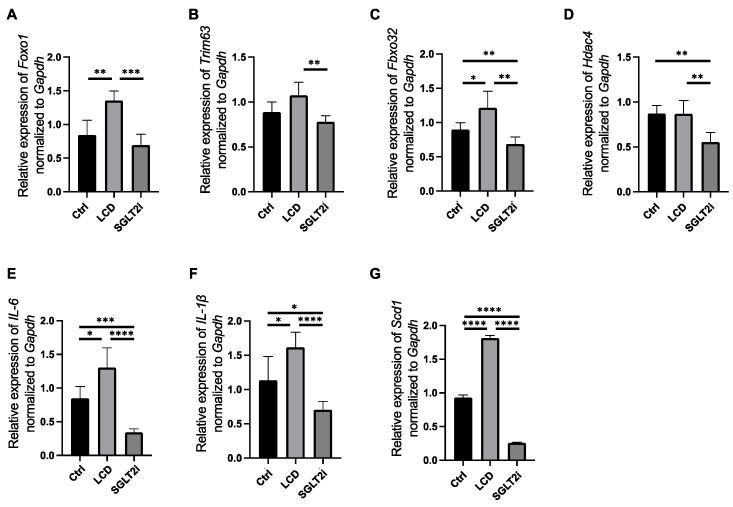
Gene expression in the soleus muscle evaluated using real-time reverse transcription polymerase chain reaction (RT-PCR). The relative expression of (**A**) Foxo1, (**B**) Trim63, (**C**) Fbxo32, (**D**) Hdac4, (**E**) Il6, (**F**) Il-1beta, and (**G**) Scd1 normalized Gapdh. Data are mean ± SD. One-way ANOVA was used for statistical analyses. * *p* < 0.05, ** *p* < 0.01, *** *p* < 0.001 and **** *p* < 0.0001.

**Figure 6 nutrients-14-03531-f006:**
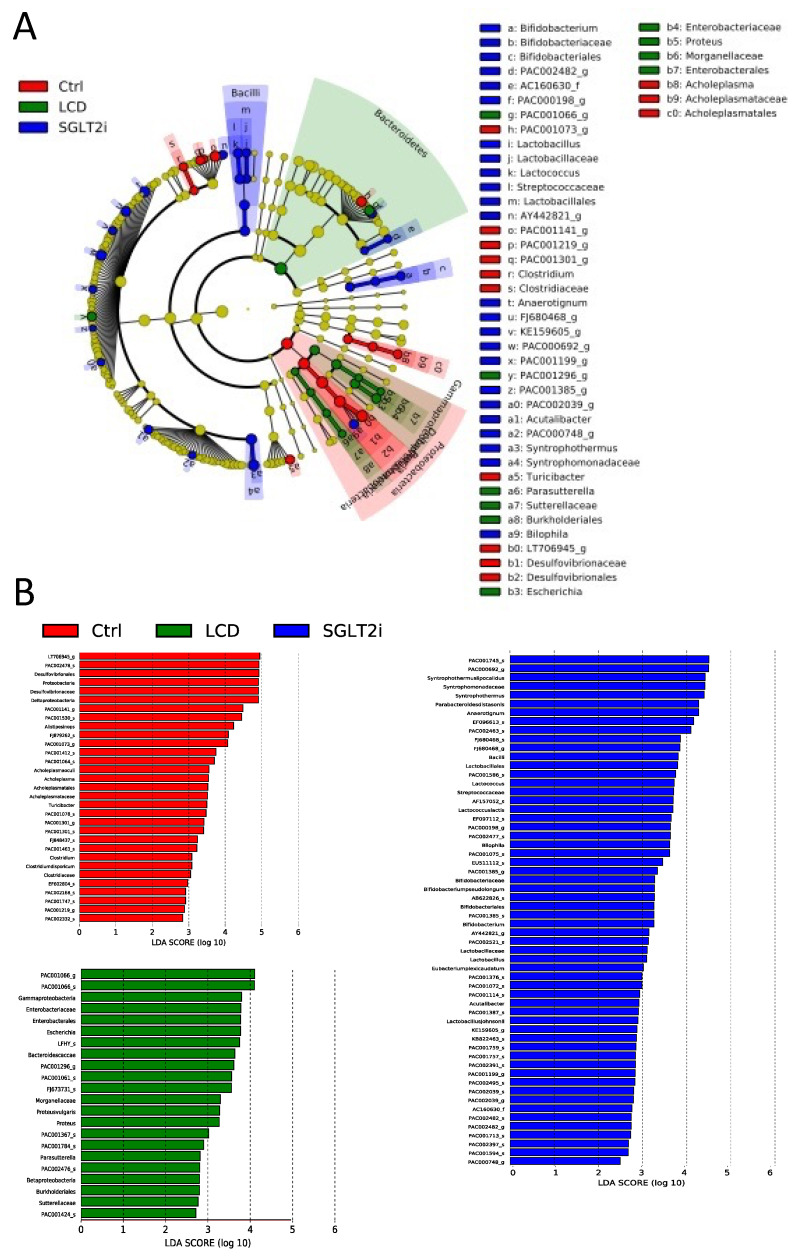
The composition of gut microbiota analyzed using shotgun metagenome analyses. (**A**) LEfSe was used to identify the taxa with the greatest differences in abundance between the gut microbiota of Ctrl, LCD, and SGLT2i groups (n = 3). Ctrl (Red); LCD (Green); SGLT2i (Blue). The brightness of each dot is proportional to the effect size. Only taxa with a significant LDA threshold value > 2 are demonstrated. (**B**) LDA scores of gut microbiota of Ctrl (Red), LCD (Green), and SGLT2i (Blue) groups.

**Table 1 nutrients-14-03531-t001:** Dietary composition for the normal diet and LCD. Abbreviation: gm, gram.

	Normal Diet			Low-Carbohydrate Diet		
	gm%	kcal%		gm%	kcal%	
Protein	20	20		28	28	
Carbohydrate	64	64		51	51	
Fat	7	16		10	21	
total		100			100	
kcal/gm	4			4		
Ingredient	gm	kcal	mg/6 gm	gm	kcal	mg/4.4 gm
Casein	200	800	1200	272.7	1090.8	1200
L-Csytine	3	12	18	4.1	16.4	18
Corn Starch	397.486	1590	2385	178.7	715	786
Maltodextrin	132	528	792	180	720	792
Sucrose	100	400	600	136	544	598
Cellulose	50	0	300	68.2	0	300
Soybean Oil	70	630	420	95.5	859.5	420
t-BHQ	0.014	0	0.084	0.019	0	0.084
Mineral Mix	35	0	210	47.7	0	210
Vitamin Mix	10	40	60	13.6	54.4	60
Choline Bitartrate	2.5	0	15	3.4	0	15
Total	1000	4000		999.969	4000	
Carbohydrate			3837			2237

## Data Availability

Data are available upon request owing to restrictions (privacy or ethical).

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
