# Peer review of "Gut Microbiota Changes by an SGLT2 Inhibitor, Luseogliflozin, Alters Metabolites Compared with Those in a Low Carbohydrate Diet in db/db Mice"

_nutrients, 2022, doi:10.3390/nu14173531_

Round 1

Reviewer 1 Report

This study aimed to compare the effect of luseogliflozin (SGLT2i) on sarcopenic obesity in comparison to that of a low-carbohydrate diet (LCD). The authors found that Luseogliflozin could prevent sarcopenic obesity by improving amino acid metabolism. The results is interesting. But there are still some major concern. The language also needs further improvement.

1. In Fig 5, The authurs detected the expression of genes involved in muscle atrophy, inflammation, and fatty acid synthesis was investigated using RT-PCR. Detection of protein level is more important and need to do the test. In the Fig 5 legend, there is an error. (F) should be il1-beta.

2. The discussion should be combined with the experimental results. Why are these detected? What is the meaning? It needs detailed discussion. As shown in Figure 5, detecting these indicators indicates what is not reflected in the discussion part.

3. In the discussion part, the authors defined that increased amino acid concentrations in the skeletal muscle of the mice in the SGLT2i group may be due to the fact that SCFAs in the gut of the SGLT2i mice activate ILC3 in the mucosal intrinsic layer of the small intestine, enhancing the mucosal protective effect of IL-22 released by ILC3. However, there is no experiment to confirm this. Additional experiments are needed to illustrate this possibility.

Author Response

Prof. Dr. Maria Luz Fernandez

Editor-in-Chief

Nutrients

                          19 April 2022

Dear Editor,

Ref.: Manuscript ID: nutrients-1874355

 Thank you for your kind letter concerning our manuscript.

Enclosed please find our revised manuscript entitled “Gut microbiota changes by an SGLT2 inhibitor, luseogliflozin, alters metabolites compared with those in a low carbohydrate diet in db/db mice”, manuscript ID of which is nutrients-1874355.

 At first, we would like to thank reviewers for constructive comments on our manuscript. According to the reviewers’ comments, we have carefully revised our manuscript. Responses to reviewers’ comments are described below. Your kind consideration of this paper would be greatly appreciated.

  We provided point-by-point responses. Please see the attachment.

 Thank you for giving us the opportunity to strengthen our manuscript with your valuable comments. We have worked hard to incorporate your feedback and hope that these revisions persuade you to accept our submission.

 We hope the revised version is now suitable for publication and look forward to hearing from you.

Yours faithfully,

Masahide Hamaguchi, MD, PhD

Department of Endocrinology and Metabolism, Kyoto Prefectural University of Medicine, Graduate School of Medical Science

Address: 465 Kajii-cho, Kawaramachi-Hirokoji, Kamigyo-ku, Kyoto 602-8566, Japan

Fax: +81-75-252-3721, Tel: +81-75-251-5505

E-mail: mhama@koto.kpu-m.ac.jp

Reviewer 2 Report

The article was well written and cohesive. I particularly missed the serum profile of the animals and also the inflammatory profile to complement the analyses.

In table 1 the authors put “gm%”. It would be interesting to describe the meaning at the end of the table.

It is important for authors to check the formatting of references and follow the instructions for authors.

In some moments in the text, the authors use the term “sacrificed”. I believe that for a bioethical issue involving animal experimentation, it is important for the authors to substitute “... euthanasia was performed”. Even in Figure 1 is one of the parts of the article that presents the word “sacrifice.”

Author Response

(The authors gave the same response as above.)

Round 2

Reviewer 1 Report

The first and third questions mentioned in the previous review need to be explained through experiments.

Reviewer 2 Report

The article may be accepted for publication.